# Signaling Pathways Impact on Induction of Corneal Epithelial-like Cells Derived from Human Wharton’s Jelly Mesenchymal Stem Cells

**DOI:** 10.3390/ijms23063078

**Published:** 2022-03-12

**Authors:** Hong Thi Nguyen, Kasem Theerakittayakorn, Sirilak Somredngan, Apichart Ngernsoungnern, Piyada Ngernsoungnern, Pishyaporn Sritangos, Mariena Ketudat-Cairns, Sumeth Imsoonthornruksa, Juthaporn Assawachananont, Nattawut Keeratibharat, Rangsirat Wongsan, Ruttachuk Rungsiwiwut, Chuti Laowtammathron, Nguyen Xuan Bui, Rangsun Parnpai

**Affiliations:** 1Embryo Technology and Stem Cell Research Center, School of Biotechnology, Institute of Agricultural Technology, Suranaree University of Technology, Nakhon Ratchasima 30000, Thailand; hongcn50c@gmail.com (H.T.N.); kasemtheera@gmail.com (K.T.); sirilak.som@ku.th (S.S.); 2Laboratory of Embryo Technology, Institute of Biotechnology, Vietnam Academy of Science and Technology, Hanoi 100000, Vietnam; 3School of Preclinical Sciences, Institute of Science, Suranaree University of Technology, Nakhon Ratchasima 30000, Thailand; apichart@sut.ac.th (A.N.); npiyada@sut.ac.th (P.N.); pishya_s@g.sut.ac.th (P.S.); 4School of Biotechnology, Institute of Agricultural Technology, Suranaree University of Technology, Nakhon Ratchasima 30000, Thailand; ketudat@sut.ac.th (M.K.-C.); tonikuya@hotmail.com (S.I.); 5School of Ophthalmology, Institute of Medicine, Suranaree University of Technology, Nakhon Ratchasima 30000, Thailand; nanont14@gmail.com; 6School of Surgery, Institute of Medicine, Suranaree University of Technology, Nakhon Ratchasima 30000, Thailand; nattawut.k@sut.ac.th; 7The Center for Scientific and Technological Equipment, Suranaree University of Technology, Nakhon Ratchasima 30000, Thailand; w_rangsirat@sut.ac.th; 8Department of Anatomy, Faculty of Medicine, Srinakharinwirot University, Bangkok 10000, Thailand; ruttachuk@g.swu.ac.th; 9Siriraj Center of Excellence for Stem Cell Research, Faculty of Medicine Siriraj Hospital, Mahidol University, Bangkok 10000, Thailand; chutila@gmail.com; 10Southeast Asia Biotechnology Center, Hanoi 100000, Vietnam; buixn5@gmail.com

**Keywords:** signaling pathways, differentiation, Wharton’s jelly mesenchymal stem cells, corneal epithelial cells, human

## Abstract

Corneal epithelium, the outmost layer of the cornea, comprises corneal epithelial cells (CECs) that are continuously renewed by limbal epithelial stem cells (LESCs). Loss or dysfunction of LESCs causes limbal stem cell deficiency (LSCD) which results in corneal epithelial integrity loss and visual impairment. To regenerate the ocular surface, transplantation of stem cell-derived CECs is necessary. Human Wharton’s jelly derived mesenchymal stem cells (WJ-MSCs) are a good candidate for cellular therapies in allogeneic transplantation. This study aimed to test the effects of treatments on three signaling pathways involved in CEC differentiation as well as examine the optimal protocol for inducing corneal epithelial differentiation of human WJ-MSCs. All-trans retinoic acid (RA, 5 or 10 µM) inhibited the Wnt signaling pathway via suppressing the translocation of β-catenin from the cytoplasm into the nucleus. SB505124 downregulated the TGF-β signaling pathway via reducing phosphorylation of Smad2. BMP4 did not increase phosphorylation of Smad1/5/8 that is involved in BMP signaling. The combination of RA, SB505124, BMP4, and EGF for the first 3 days of differentiation followed by supplementing hormonal epidermal medium for an additional 6 days could generate corneal epithelial-like cells that expressed a CEC specific marker CK12. This study reveals that WJ-MSCs have the potential to transdifferentiate into CECs which would be beneficial for further applications in LSCD treatment therapy.

## 1. Introduction

Cornea, the anterior transparent part of the eye, permits light transmission to photoreceptor cells in the retina and protects the eye from the external environment. The cornea consists of three cell layers: corneal epithelium, stroma, and endothelium. The corneal epithelium, the outmost layer of the cornea, comprises corneal epithelial cells (CECs) that are continuously renewed by limbal epithelial stem cells (LESCs). LESCs are located at the basal layer of the limbus [1] which is the border between the cornea and the sclera. LESCs can divide asymmetrically to produce both LESC daughters and transit-amplifying cells. While LESCs remain in the limbus, transit-amplifying cells migrate into the central cornea and move upward to the superficial layer of the cornea to differentiate into CECs [2,3]. Loss or dysfunction of LESCs due to several types of damages (chemical or thermal burns), microbial infections, diseases such as Stevens–Johnson syndrome can result in limbal stem cell deficiency (LSCD) [4,5,6]. LSCD leads to loss of corneal epithelial integrity and function, resulting in vision loss or corneal blindness [7,8].

Common therapeutic treatments of LSCD include cultivated limbal epithelial transplantation (CLET) and cultivated oral mucosal epithelial transplantation (COMET). However, autologous CLET is impossible in the case of bilateral LSCD, and allogenic CLET requires the long-term use of systemic immunosuppression [9]. COMET gives promising results for the stabilization of the ocular surface [10], but the transplanted oral cells did not fully transdifferentiate into CECs (did not express CK12, a specific marker of CECs) [11]. Both techniques give variation in success rate, use animal feeder cells that has risks of contamination or disease transmission from other species, and cause peripheral corneal neovascularization [12,13,14].

To solve these problems, researchers are trying to find new cell sources that are better candidates for transplantation to treat LSCD, such as embryonic stem cells (ESCs), induced pluripotent stem cells (iPSCs), mesenchymal stem cells (MSCs) from Wharton’s Jelly (WJ) or dental pulp, etc. ESCs are pluripotent stem cells but research on human ESCs is ethically and politically controversial because of its involvement in the destruction of human embryos [15]. Human iPSCs have the same differentiation capacity as human ESCs and even avoid post-transplantation rejection by using the patient’s own somatic cells. However, the factors associated with iPSCs generation have been linked to oncogenic transformation, a form of in vitro produced tumor cells [16]. MSCs have lower differentiation potential but they are safer than iPSCs. Transplantation of human MSCs was successfully used for the treatment of ocular surface disorders (such as congenital corneal diseases of genetic mutation [17], corneal defects of Mucopolysaccharidosis VII [18], and LSCD [19,20], etc.). However, the in vivo transdifferentiation of MSCs into CECs remains unclear [20,21]. WJ-MSCs are good candidates for cellular therapies in allogenic transplantation due to their capacity for immune suppression and immune avoidance [22]. Human WJ-MSCs could differentiate into CECs by culturing on artificial corneal stroma which was generated using human keratocytes [23]. However, the optimal protocols using defined medium for generating CECs from WJ-MSCs in vitro have not been reported yet. 

Both condition medium and defined induction medium were used to differentiate pluripotent stem cells into CECs (reviewed in [24]). In methods using defined induction medium, several combinations of treatment factors that functioned in inhibiting the Wnt signaling pathway (retinoic acid, IWP2, IWR1), upregulating the TGF-β signaling pathway (SB505124, A83-01) with/without increasing the BMP signaling pathway (BMP4) had succeeded in generating CECs from iPSCs and ESCs [10,25,26]. However, there are few studies evaluating the effects of treatment factors on these signaling pathways, especially with WJ-MSCs. Therefore, this study aimed to determine the effects of treatment factors (all-trans retinoic acid (RA), SB505124, and BMP4) on Wnt, BMP, and TGF-β signaling pathways and investigate the optimal protocol for generating CECs from human WJ-MSCs.

## 2. Results

### 2.1. Human WJ-MSCs Characterization

Primary human WJ-MSCs were successfully isolated by explant culture. Cells were plastic-adherent and showed fibroblast-like morphology (Figure 1A). The flow cytometry result indicated that >95% of the cell population expressed standard markers of MSCs (positive rates of CD90, CD73, CD105 were 99.78%, 99.54%, 99.03%, respectively, Figure 1B) and <2% cell population expressed hematopoietic markers (positive rates of CD34, CD45 were 0.12%, 0.55%, respectively, Figure 1B). Multipotency was determined by trilineage differentiation protocols. After adipogenic differentiation, cells produced lipid droplets stained with Oil Red O (Figure 1C, left). Calcified matrix deposition was confirmed with Alizarin Red staining (Figure 1C, middle) after osteogenic differentiation. Additionally, glycosaminoglycans were stained with Alcian blue (Figure 1C, right) after treatment with chondrogenic medium. The self-renewal capacity of human WJ-MSCs was demonstrated by PDT analysis. PDT of the cells at passage 3 (P3) was similar with the cells at P4 (27.22 ± 0.82 h vs. 28.88 ± 0.60 h; *p* > 0.05), but lower than the cells at P5 to P10 (*p* < 0.05) (Figure 1D). PDTs of cells at P5 to P10 were not significantly different (average 37.37 h; *p* > 0.05).

### 2.2. Cytotoxicity

Cytotoxicity effects of RA, SB505124, and DMSO on the viability of human WJ-MSCs are shown in Figure 2A. DMSO treatment was used as vehicle control. DMSO caused a concentration-dependent decrease in cell viability. DMSO at dose 0.1% did not significantly reduce cell viability (*p* > 0.05). However, DMSO at concentrations of 0.25%, 0.5%, and 1% significantly decreased cell viability compared with control without DMSO (92.31 ± 1.43%, 87.53 ± 1.51%, and 85.02 ± 2.08% vs. 100.00 ± 1.69%, respectively; *p* < 0.05). RA also showed cytotoxicity with WJ-MSCs. The survival rates of WJ-MSCs following treatment with RA 2.5, 5, 10, 20 µM were significantly lower than control (77.83 ± 4.38%, 84.45 ± 2.03%, 81.43 ± 2.08%, and 80.03 ± 1.69% vs. 100 ± 1.69%; *p* < 0.001). Unlike RA and DMSO, SB505124 was not cytotoxic. Treatment with SB505124 showed a trend increased in the viability of WJ-MSC but was not significant compared to control.

### 2.3. Effect of RA on Localization and Expression of β-Catenin

Human WJ-MSCs retained fibroblast-like morphology in control while RA treatment caused the cells to become more flattened and shorter in length (Appendix A). RA treatment had no effect on total β-catenin protein expression (Figure 2D,E) and mRNA expression (Figure 2F). However, the ratio of mean fluorescence intensity of β-catenin in the nucleus to the cytoplasm was significantly suppressed in both RA groups (5 and 10 µM) compared with the control group (0.88, 0.97 vs. 1.15; *p* < 0.05) (Figure 2C). RA 5 and 10 µM inhibited β-catenin translocated from the cytoplasm into the nucleus. RA treatment did not affect mRNA and protein expression of β-catenin but both concentrations of RA (5 and 10 µM) could suppress the Wnt/β-catenin signaling pathway via reducing translocation of β-catenin from the cytoplasm into the nucleus.

### 2.4. Effect of SB505124 on Inhibition of p-Smad2/3

Western blot results of SB505124 treatment are shown in Figure 3A. Ratio of p-Smad2/Smad2 protein intensity was significantly reduced after treatment with all concentrations of SB505124 (Figure 3C, left). Phosphorylation of Smad3 tended to decrease after treatment with SB505124 (Figure 3C, right). However, only 20 µM SB505124 was significantly suppressed on phosphorylation of Smad3. 

### 2.5. Effect of BMP4 for Increasing p-Smad1/5/8

Western blot results of SB505124 treatment are shown in Figure 3B. Treatment with BMP4 for 1 h did not affect phosphorylation of Smad1/5/8 (Figure 3D). The ratio of p-Smad1/5/8/Smad1/5/8/9 was not different after 2 h of 50 ng/mL BMP4 treatment. Treatment with 25 ng/mL BMP4 for both 1 and 2 h did not increase phosphorylation of Smad1/5/8. Both concentrations of BMP4 had no significant effect on the phosphorylation of Smad1/5/8 proteins, indicating no activation of the BMP signaling pathway.

### 2.6. Characterization of Human CECs

Isolated human CECs showed characteristics of normal CECs such as corneal specific marker (CK12) and other markers of CECs (E-cadherin, ZO-1, Involucrin). Human CEC morphology is shown in Figure 4A. Cells looked like cobblestones, varying in size and shape. Cells at P1 stained positive with CK12 and E-cadherin. ZO-1 and Involucrin stained in the large CECs.

### 2.7. Differentiation of Human WJ-MSCs into CECs

During differentiation, the morphology of cells was changed in all groups (Figure 4B). After 9 days, cells became larger and flatter. Especially the morphology of differentiated cells in group G2 at day 9 and groups G1, G2, G3 at day 12 looked similar with epithelial-like cells. CK12 protein (specific marker of corneal epithelial cells) expression is shown in Figure 5A. In the BM group, there were rarely cells stained with CK12. However, in groups G1, G2, and G3, the positive CK12 cell number was increased from day 3 to day 9 then reduced at day 12. CK12 expressed highest in group G2 at day 9 after differentiation. Although, levels of both CK12 and CK3 (specific CEC markers) mRNA expression in G2 at day 9 were lower than in cornea, they were significantly higher in control (*p* < 0.05; Figure 5B). ATP-binding cassette transporter (ABCG2, a marker of putative LESCs) protein expression is shown in Figure 6A. Almost cells in the control and treatment groups showed expression of ABCG2. However, the expression of ABCG2 in the control group was very low. After treatment, ABCG2 expression was higher in all groups than the control group. ABCG2 expression at day 12 was lower than day 3 and day 6 in all groups. Level of ABCG2 mRNA expression tended to increase in all treatment groups compared with the control group but there was no significant difference between them (Figure 6B, left). Moreover, other corneal epithelial progenitor markers (CK15 and p63) significantly increased mRNA expression in group G2 at day 9 compared with the control group (*p* < 0.05; Figure 6B and Figure 7B, right). Additionally, Paired Box 6 (PAX6, an essential transcription factor for development and function of the cornea) was also upregulated in group G2 at day 9 compared with control groups (Appendix A). CK19 (a marker of conjunctival epithelial marker) staining is shown in Figure 7A. A subpopulation of cells in the control group stained strong positively with CK19. After treatment, the intensity of CK19 was reduced in all treatment groups. Moreover, level of CK19 mRNA showed a decrease in all treatment groups compared with the control group. Level of CK19 mRNA in the cornea was lower than control WJ-MSCs but higher than all treatment groups (*p* < 0.001). All results indicated that the treatment condition of group G2 using a combination of RA, SB505124, BMP4, EGF in the 1st step could generate corneal epithelial cells from human WJ-MSCs with the highest efficiency after 9 days.

## 3. Discussion

MSCs are one of the most common cell types that are used for regenerative medicine [27]. Especially, WJ-MSCs are good candidates for cellular therapies in allogenic transplantation due to their immune suppression and immune avoidance capacity [22,28]. Moreover, WJ-MSCs are more easily isolated and have higher proliferation potential compared with MSCs from adipose tissue, cord blood, placenta, and bone marrow [28]. In this study, we successfully isolated and expanded human WJ-MSCs that fit the minimal criteria characterizing human MSCs [29] such as (i) plastic-adherent; (ii) express CD105, CD73, CD90 and lack expression of CD34, CD45; (iii) can differentiate to osteoblasts, adipocytes, and chondroblasts. Moreover, similar to the reported WJ-MSCs [28,30], our isolated WJ-MSCs were also fibroblast-like cells and showed high potency of self-renewal capacity. Cell at passages 3 and 4 had higher self-renewal capacity than cells in passages 5–10.

Although transplantation of MSCs successfully reconstructed the damaged corneal surfaces of rats [20], mice [31], rabbits [32], and humans [19], the therapeutic effectiveness of MSC transplantation may be caused by their suppression of inflammation and angiogenesis rather than the epithelial transdifferentiation [20,21]. Rat cornea transplanted with MSCs did not express CK3, CK12 [20,33]. These results indicated that the transdifferentiation potential of transplanted MSCs in an in vivo model was uncertain. Other studies focused on finding methods of generating CECs from MSCs in vitro. These methods were based on co-culture with LESCs [34], CECs [35,36], or conditioned medium from limbal explant [37]. These methods are needed to culture signal providing cells that had risks of contamination or disease transmission. The medium compositions used in these methods were undefined and uncontrollable. Moreover, the co-culture system required expensive equipment. Other research used defined media to induce CECs derived from conjunctiva-MSCs, BM-MSCs [35,38,39,40]. In this study, we focused on finding an optimal method to differentiate WJ-MSCs into CECs in vitro by comparing three combinations in the first step and differentiation duration (9 or 12 days). We found that the combination (RA, SB505124, BMP4, EGF) is the best, and the differentiation time is 9 days. After differentiation, cells were positively stained with the specific marker of CECs (CK12) and mRNA expression of both CK3 and CK12 was upregulated. 

The Wnt/β-catenin signaling pathway plays a vital role during the proliferation of LESCs [41]. During normal homeostasis of the corneal epithelium, Wnt/β-catenin signaling may be relatively inactive and β-catenin is mainly membrane-bound in the normal intact corneal epithelium [41,42]. Inhibiting Wnt signaling results in differentiation into corneal epithelial cells [10,43]. In this study, RA (5 and 10 µM) treatment could suppress the Wnt/beta-signaling pathway via inhibiting translocation of β-catenin from cell cytoplasm into the nucleus. Lower concentration of RA (1 µM) induced membrane localization of β-catenin and downregulated expression of β-catenin in the nucleus of human ESCs [43]. However, 1 µM RA did not only suppress the canonical Wnt signaling pathway but also activate the noncanonical Wnt signaling pathway in murine ESCs [44]. Therefore, in this study, 10 µM RA was used for inducing WJ-MSCs differentiation into CECs. Furthermore, TGF-β signaling pathway also regulates epithelial differentiation in eye development [45]. Suppression of TGF-β signaling is necessary for generating CECs from human iPSCs [10]. SB505124 is one of the selective inhibitors of the activin and TGF-β signaling pathway [46]. In this study, SB505124 treatment inhibited phosphorylation of the cytoplasmic signal transducer (Smad2) of the TGF-β signaling pathway, so SB505124 could inhibit this signaling pathway. In previous studies, BMP4 combined with suppressing Wnt/β-catenin signaling together with/without inhibiting TGF-β signaling had an effect on CEC differentiation from human iPSCs [10] and BM-MSCs [39]. In this study, BMP4 (25 or 50 ng/mL) supplementation did not significantly improve phosphorylation of the transducer (Smad1/5/8). Another signaling pathway, bFGF, was necessary for generating CECs from human iPSCs [10,25]. However, supplementation of bFGF did not improve CEC differentiation from human WJ-MSCs in this study. This result may be caused by the presence of BMP4 in the treatment groups. Similar to bFGF, BMP4 (10 ng/mL) upregulated phosphorylation of extracellular signal-related kinases (ERK1/2) in human CECs [47]. In this study, the combination of Wnt and TGF-β signaling inhibitors together with BMP4 and EGF supplementation could generate corneal epithelial cells from human WJ-MSCs with the highest efficiency compared to other treatment combinations. After 9 days of differentiation, induced cells expressed a specific protein of CECs (CK12) and mRNA expression of specific markers (CK12, CK3) was upregulated. Increasing expression of both CK3 and CK12 was also reported in CEC derived from human iPSCs [10,26], ESCs [25], conjunctiva-MSCs [35,38], BM-MSCs [40], WJ-MSCs [23], and rabbit adipose tissue-MSCs [37].

ATP-binding cassette transporter (ABCG2), breast cancer resistance protein 1 (BCRP1), is considered as a marker for many stem cell lines [48]. ABCG2 was identified as a marker of putative LESCs [49,50,51]. ABCG2 was also shown positive with human umbilical cord matrix stem cells (hUCMS) [52], human dental pulp-MSC [32], and rat BM-MSCs [53]. In this study, human WJ-MSCs showed low expression of ABCG2 but the expression of ABCG2 was upregulated after differentiation. Increasing ABCG2 gene expression was also indicated in corneal epithelial-like cells derived from human iPSCs [26], rabbit adipose tissue-MSCs [37]. Together with ABCG2, CK15 and p63 are also putative markers of LESCs [50,51,54]. Upregulation of CK15 and p63 genes of induced cells in this study was similar with previous CECs derived from human iPSCs [10], ESCs [25], and rabbit adipose tissue-MSCs [37]. Moreover, increasing gene expression of PAX6, a coactivator of the CK12 gene [55], in this study was similar to the previous studies [25,26,37]. Another marker of LESCs, CK19, was shown in conjunctival epithelial cells and peripheral corneal basal cells [56]. CK19 expression was shown in the subpopulation of chorionic-plate-MSCs, chorionic villi-MSCs, and WJ-MSCs [30]. In our study, some human WJ-MSCs also stained positive for CK19. However, CK19 gene expression was downregulated after differentiation in this study. Reduced expression of CK19 also was observed in CECs derived from BM-MSCs [40].

## 4. Materials and Methods

### 4.1. Reagents

All chemical compounds and cell culture reagents were purchased from Sigma-Aldrich Corporation (St. Louis, MO, USA) and Thermo Fisher Scientific (Waltham, MA, USA), respectively, and cell culture ware was obtained from SPL Life Science (Gyeonggi-do, Korea), unless stated otherwise.

### 4.2. Isolation and Expansion of Human WJ-MSCs

The human umbilical cord was collected from Maharat Nakhon Ratchasima Hospital (Nakhon Ratchasima, Thailand) after the mother’s informed consent was obtained. Human WJ-MSCs were isolated from the umbilical cord and cultured as previously described [57,58]. Briefly, the umbilical cord was put into 75% ethanol for 30 s, 10% betadine, and washed in sterilized PBS. Then, the umbilical cord was cut lengthwise and the arteries and veins were removed. The gelatinous WJ tissue was excised and cut into small fragments (3 × 3 mm). WJ fragments were plated into 60 mm dishes and covered with 4 mL of αMEM supplemented with 10% fetal bovine serum (FBS), 100 U/mL penicillin, 100 µg/mL streptomycin. WJ fragments were incubated at 37 °C in a humidified atmosphere of 5% CO_2_ for 10–13 days. The culture medium was replaced every 3 days. When the visible colonies were observed, cells were sub-cultured into T75 flasks at the density of 10^4^ cells/cm^2^. The cells were expanded until passage 3 (P3), then the cells were either directly used for experiments (sub-culture to P4) or cryopreserved in culture media supplemented with 10% dimethyl sulfoxide (DMSO, Calbiochem, San Diego, CA, USA) and stored in liquid nitrogen.

### 4.3. Flow Cytometry

Human WJ-MSCs were harvested and washed with PBS. Afterwards, approximately 2 × 10^5^ cells were suspended in a final volume of 100 µL PBS and incubated with primary antibodies (CD73-APC, CD90-APC/Cy7, CD105/PE, CD34-PE, and CD45/FITC) for 20 min at room temperature, in the dark. As negative controls, isotype control antibodies were used. The cells were washed and resuspended in a final volume of 500 µL PBS. At least 10^4^ cells were determined on an Attune^TM^ NxT Flow Cytometer (Thermo Fisher Scientific). Finally, the data obtained were analyzed using FlowJo^TM^ v10.8 Software (BD Life Sciences, Ashland, OR, USA). The details of primary antibodies are shown in Appendix A. 

### 4.4. Trilineage Differentiation Capacity

Trilineage differentiation capacity of human WJ-MSCs was evaluated as previously described [58]. Cells were cultured in 35 mm at the density of 2 × 10^3^ cells/cm^2^ for 2–3 days. Cells treated with the adipogenic medium were stained with Oil Red O after 21 days. Additionally, chondrogenic media-treated cells were stained with Alcian Blue on day 21. Moreover, cells treated with osteogenic medium for 21 days were stained with Alizarin Red. Adipogenic medium comprised αMEM supplemented with 5% FBS, 10 µg/mL insulin, 10 µM indomethacin, 1 µM dexamethasone (DEX), 0.5 mM isobutylmethylxanthine (IBMX). Chondrogenic medium contained αMEM supplemented with 2% FBS, 1% Insulin-Transferrin-Selenium-Ethanolamine (ITS-X), 50 µg/mL ascorbate-2-phosphate (A2P), 40 µL L-proline, 100 µg/mL sodium pyruvate, 100 nM DEX, and 10 ng/mL of TGF-β3 (Prospec, East Brunswick, NJ, USA). Osteogenic medium consisted of αMEM supplemented with 5% FBS, 100 nM DEX, 0.2 mM A2P, and 10 mM β-glycerophosphate. The medium was changed every three days.

### 4.5. Cytotoxicity Test

Cells were seeded at a density of 3000 cells/well in 96-well culture plates in the culture medium for 24 h. Then, the cells were treated with culture medium supplemented with several concentrations of SB505124 (0, 5, 10, and 20 µM) or DMSO (0, 0.1, 0.25, 0.5, 1%) or RA (0, 2.5, 5, 10 and 20 µM) at 37 °C for 72 h in a humidified atmosphere of 5% CO_2_ in air. After treatment, cell viability was quantified by MTT assay as previously described [58]. Briefly, cells were incubated with culture medium supplemented with 0.5 mg/mL 3-(4,5-Dimethylthiazol-2-yl)-2,5-Diphenyltetrazolium Bromide (MTT, Invitrogen) for 3 h at 37 °C. Then, 0.01 M DMSO was added, and the cells were incubated for 10 min at 37 °C. The absorbance at 540 nm was read by Thermo Scientific^TM^ Multiskan^TM^ GO Microplate Spectrophotometer (Thermo Fisher Scientific, Waltham, MA, USA). Each treatment condition was performed in 4 replicates.

### 4.6. Population Doubling Time (PDT)

Cells at passages 3–10 were plated in triplicate onto the 12-well plate at a density of 5000 cells/cm^2^ and cultured in αMEM supplemented with 10% FBS. After 3 days, the numbers of viable cells were counted using Trypan blue staining. PDT was calculated using the following formula: PDT = (CT × ln2)/ln(Nf/Ni), where CT is the cell culture time (hours), Ni and Nf are the initial and the final numbers of cells, respectively [59].

### 4.7. Effect of RA on Localization and Expression of β-Catenin

Cells were seeded at a density of 10^3^ cells/cm^2^ and incubated for 48 h. They were then treated with basic medium (BM: DMEM low glucose supplemented with 2% FBS, 1% NEAA, 100 U/mL penicillin, 100 µg/mL streptomycin) supplemented with RA (0, 2.5, 5, 10 µM) for 3 additional days at 37 °C and 5% CO_2_. The expression β-catenin was analyzed by immunofluorescent staining, Western blot, and qPCR.

### 4.8. Effect of SB505124 on Inhibition of p-Smad2/3

Cells were seeded at a density of 10^3^ cells/cm^2^ and cultured in the culture medium for 3 days. Then cells were cultured in DMEM/F12 medium without FBS for 24 h. Afterwards, cells were treated with SB505124 (0, 5, 10, and 20 µM) in DMEM/F12 for 1 h. The expressions of total Smad2/3, pSmad2/3, β-actin were evaluated by Western blot.

### 4.9. Effect of BMP4 for Increasing p-Smad1/5/8

Cells were seeded at a density of 10^3^ cells/cm^2^ and cultured in the culture medium for 3 days. Then they were cultured in DMEM/F12 medium without FBS for 24 h. Afterwards, they were treated with BMP4 (0, 25, and 50 ng/mL) in BM for 1 or 2 h. The expression of total Smad1/5/8/9, pSmad1/5/8, and β-actin was evaluated by Western blot.

### 4.10. Isolation and Characterization of Human CECs

Human cadaveric limbal tissue consisting of peripheral cornea and limbus was obtained from Eye Bank of Thailand (The Thai Red Cross Society, Bangkok, Thailand) and stored in Optisol GS (Bausch & Lomb, Rochester, NY, USA) at 4 °C. The endothelial layer and iris remnants were removed, and the cornea was then cut into small fragments (2 × 2 mm). These fragments were used for mRNA isolation or cultured in supplemented hormonal epidermal medium (SHEM) containing a mixture of DMEM low glucose and DMEM/F12 medium (1:1 *v/v*) supplemented with 5% FBS, 10 ng/mL EGF, 1% of insulin-transferrin-sodium selenite (ITS-H, Capricorn Scientific GmbH, Ebsdorfergrund, Germany), 0.5 µg/mL hydrocortisone, 0.05% DMSO, 200 nM adenine, 100 U/mL penicillin, 100 µg/mL streptomycin in 37 °C and 5% CO_2_. After cell proliferation, the fragments were removed, and the cells were cultured for an additional week. The culture medium was changed every 3 days. When the cells reached 70–80% confluence, cells were sub-cultured and seeded at density 3 × 10^4^ cells/cm^2^. Cells were then characterized by immunofluorescent staining for CK12, E-cadherin (an intercellular junction protein), zonula occludens-1 (ZO-1, a tight junction protein), and Involucrin.

### 4.11. Optimization of Human WJ-MSC Differentiation into CECs

Human WJ-MSCs were seeded at a density of 10^3^ cells/cm^2^ and cultured in αMEM supplemented with 10% FBS for 2 days. Then, these cells were treated with BM supplemented with or without combinations of treatment factors (G1: RA + SB505124 + BMP4 + bFGF + EGF; G2: RA + SB505124 + BMP4 + EGF; G3: RA + BMP4 +bFGF + EGF) for 3 days. Concentrations of these factors were 10 µM RA, 10 µM SB505124, 25 ng/mL BMP4, 50 ng/mL bFGF, 10 ng/mL EGF. Afterwards, these cells were cultured in SHEM medium for an additional 6 or 9 days. These cells at days 0, 3, 9, and 12 were evaluated by immunofluorescent staining, Western blot, and qPCR. A schematic outline of the CEC differentiation process is shown in Figure 8.

### 4.12. Immunofluorescent Staining

Expression of β-catenin, CK12, CK19, ABCG2 was qualitatively evaluated with immunofluorescent staining. Cells were fixed in cold absolute methanol for 20 min and washed three times with PBS. Cell membranes were permeabilized for 30 min in 0.2% Triton X-100. Nonspecific binding sites were blocked with 1% BSA in PBS for 1 h. Cells were then stained with primary antibodies diluted with 1% BSA/PBS for 1 h at room temperature (RT) or overnight at 4 °C. Afterwards, cells were stained with secondary antibodies diluted in PBS for 1 h at RT. Nuclei were stained with 1 µg/mL 4, 6-diamino-2-phenylindole (DAPI, Millipore) for 5 min. Then, the cells were mounted with Vectashield^®^ antifade mounting medium (H-1000, Vector Laboratories, Burlingame, CA, USA). Cell images were captured with a Nikon Eclipse Ti-S fluorescent microscope equipped with a DS-Ri1 camera (Nikon Instruments Inc., Tokyo, Japan). The intensity of β-catenin in the nucleus and cytoplasm was measured using CellProfiler software (www.cellprofiler.org, accessed on 12 January 2022).

### 4.13. Western Blotting Analysis

Cell samples were lysed in lysate buffer (10 mM Tris-HCl, 150 mM NaCl, 0.5% Triton X-100, 1 mM EDTA, pH 7.2) supplemented with protease inhibitor cocktail (cOmplete^TM^) and phosphatase inhibitor cocktail (PhosSTOP^TM^). The cell lysates were cleared by centrifugation at 14,000 rpm for 20 min at 4 °C, then, the total protein concentration was determined by Bradford assay (Coomassie protein assay reagent). An equal amount of total protein (10–20 µg) from each sample was mixed with 5× Laemmli buffer before denaturing at 95 °C for 5 min and separated on 7.5% or 10% Acrylamide/Bis gels. Afterwards, separated proteins were transferred onto PVDF membranes (Immu-Blot PVDF membrane, Bio-Rad, Hercules, CA, USA). Nonspecific binding sites were blocked by blocking buffer (5% skim milk in Tris-buffered saline supplemented with 0.1% Tween-20 (TBST)) for 1 h at RT. The membranes were then incubated with primary antibodies diluted in 1% BSA in PBS at 4 °C overnight. After washing in TBST, membranes were incubated with secondary antibody conjugated to horseradish peroxidase (HRP) diluted 1:5000 in 5% skim milk in TBST at RT for 1 h and then developed by using a ECL substrate kit (Ultra high sensitivity, Abcam). Protein bands were imaged by ImageQuant LAS 500 (GE Healthcare Life Sciences, Piscataway, NJ, USA) and then quantified using Image J. Details of 1st and 2nd antibodies are listed in Appendix A. β-actin was used as a protein loading control.

### 4.14. Real-Time Quantitative PCR (qPCR)

Total mRNA was extracted from samples by using FavorPrep^TM^ Tissue total RNA mini kit (Favorgen Biotech corp., Taipei, Taiwan). RNA concentration of each sample was measured using a NanoDrop Spectrophotometer (NanoDrop Technologies, Wilmingtion, DE, USA). From each RNA sample, 500 ng was used to synthesize first-strand cDNA using a cDNA synthesis kit (Biotech rabbit, Berlin, Germany). Then, qPCR reactions were carried out with cDNA, KAPA SYBR^®^FAST qPCR kit (KAPA Biosystems, Woburn, MA, USA), primers shown in Appendix A, and the amplifications were performed in QuantStudio^TM^ 5 real-time PCR system (Thermo Fisher Scientific). Results were analyzed with QuanStudio^TM^ Design and Analysis and Microsoft Excel software. Melting curve analysis was used to confirm the specificity of the primers. The relative quantification of each gene was calculated by applying the −2^∆∆Ct^ method [60]. Results were normalized to GAPDH with undifferentiated WJ-MSCs as the calibrator to determine the relative quantities of gene expression in each sample. All samples and controls were run as triplicate reactions.

### 4.15. Statistical Analysis

All experiments were repeated three times. All data were presented as mean ± standard error of the mean (SEM) from three separate experiments. All statistical analyses were carried out in SAS^®^ Studio (SAS Institute, Cary, NC, USA), using one-way ANOVA, and Duncan’s multiple range test was used as a post hoc test. *p* < 0.05, *p* < 0.001 were considered statistically significant.

## 5. Conclusions

In summary, this study described the effects of treatment factors (RA, SB505124, and BMP4) on the involved signaling pathways in human WJ-MSCs, then compared several combinations of these treatment factors on the differentiation of these cells into CECs. RA inhibits Wnt signaling via reducing translocation of β-catenin while SB505124 suppresses TGF-β signaling by decreasing phosphorylation of Smad2. This study indicates a feeder-free, non-conditioned medium 2-step method to generate CECs from WJ-MSCs within 9 days. This differentiation method consists of two steps: the first step uses a combination of RA, SB505124, BMP4, and EGF and the second step uses SHEM medium. Induced CECs derived WJ-MSCs are valuable for research studies on LSCD treatment in an in vivo model.

## Figures and Tables

**Figure 1 ijms-23-03078-f001:**
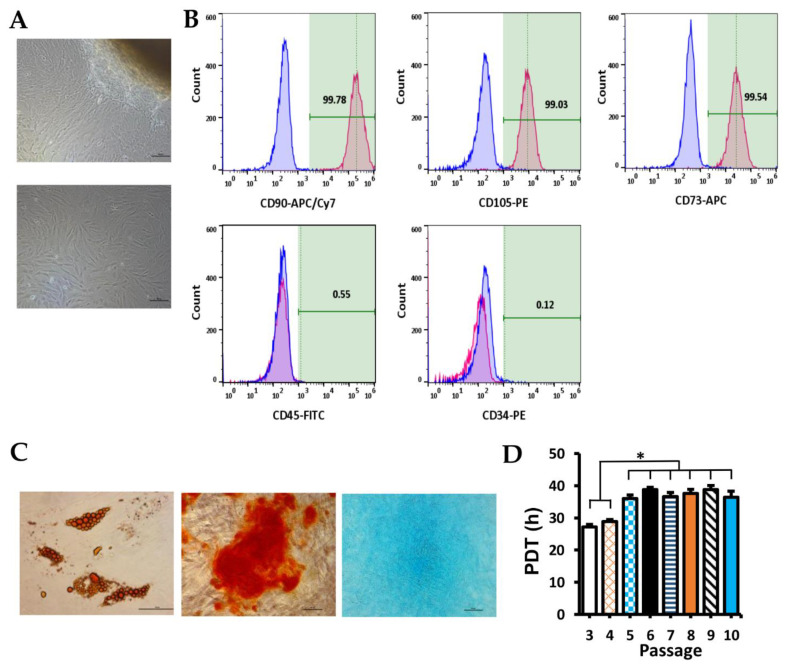
Characterization of human WJ-MSCs. (**A**) Figure of explant culture of WJ tissue at day 8 (upper) and WJ-MSCs of P1 day 3 (lower); scale bar, 100 µm. (**B**) Flow cytometry results with MSC markers (CD90, CD73, CD105) and hematopoietic markers (CD34, CD45). (**C**) Trilineage differentiated cells with Oil Red O staining (left), Alizarin Red staining (middle), and Alcian Blue staining (right); scale bar, 50 µm. (**D**) PDT at different passages (from 3 to 10). Data are presented as mean + SEM. * *p* < 0.05.

**Figure 2 ijms-23-03078-f002:**
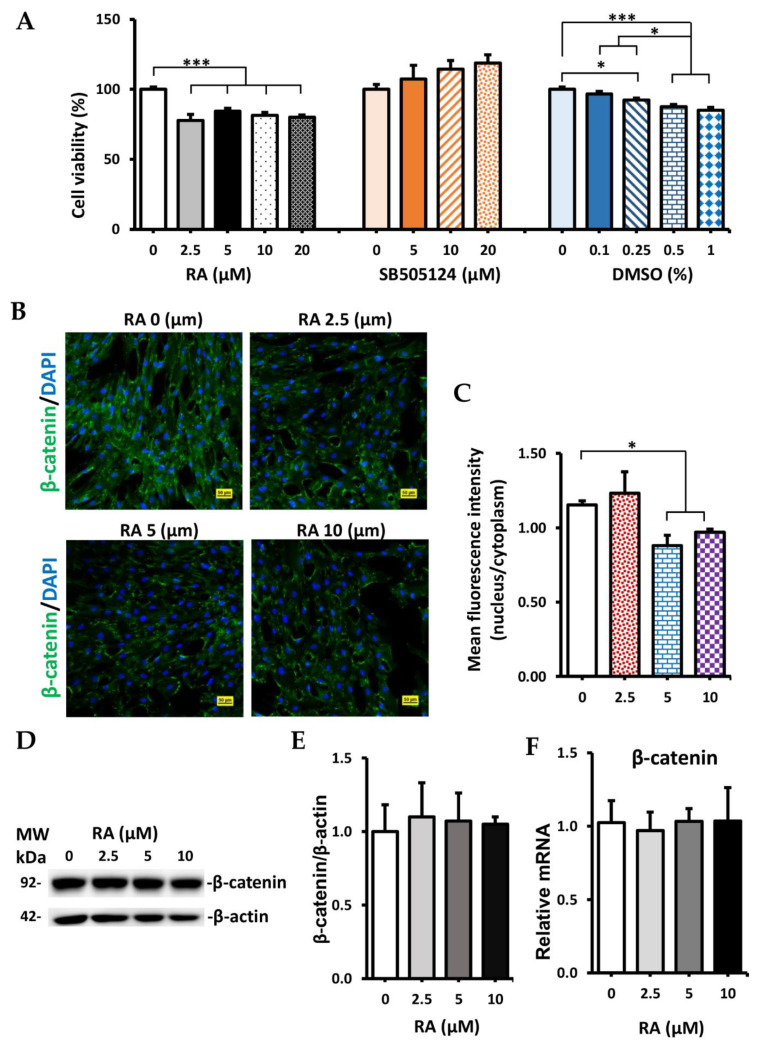
Cytotoxicity of treatment factors and effect of RA on β-catenin expression. (**A**) Cytotoxicity of RA, SB505124, and DMSO on WJ-MSCs. (**B**) Immunostaining of WJ-MSCs with β-catenin (green) after treatment with RA. Scale bar, 50 µm. (**C**) Ratio of mean fluorescence intensity of β-catenin in the nucleus to cytoplasm was analyzed by confocal microscopy. (**D**) Western blot images of β-catenin, β-actin expression. (**E**) Quantification of Western blot results. (**F**) The relative mRNA expression of β-catenin. Data are presented as mean + SEM. * *p* < 0.05, *** *p* < 0.001.

**Figure 3 ijms-23-03078-f003:**
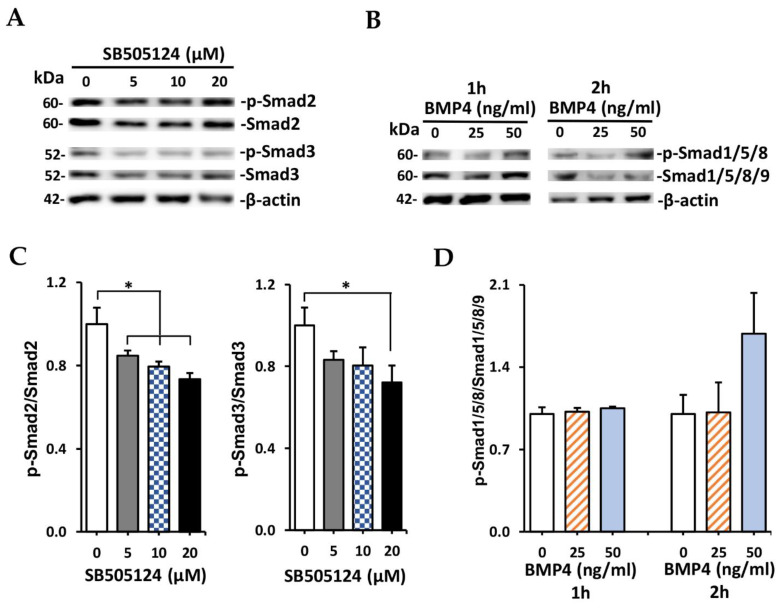
Effect of SB505124, BMP4 on phosphorylation of Smad2/3, Smad1/5/8 (respectively). (**A**) Western blot images of p-Smad2/3, Smad2/3, β-actin expression. (**B**) Western blot images of p-Smad1/5/8, Smad1/5/8/9, β-actin expression. (**C**) Quantification of Western blot results of p-Smad2 (left) and p-Smad3 (right). (**D**) Quantification of Western blot results of p-Smad1/5/8. Data are presented as mean + SEM. * *p* < 0.05.

**Figure 4 ijms-23-03078-f004:**
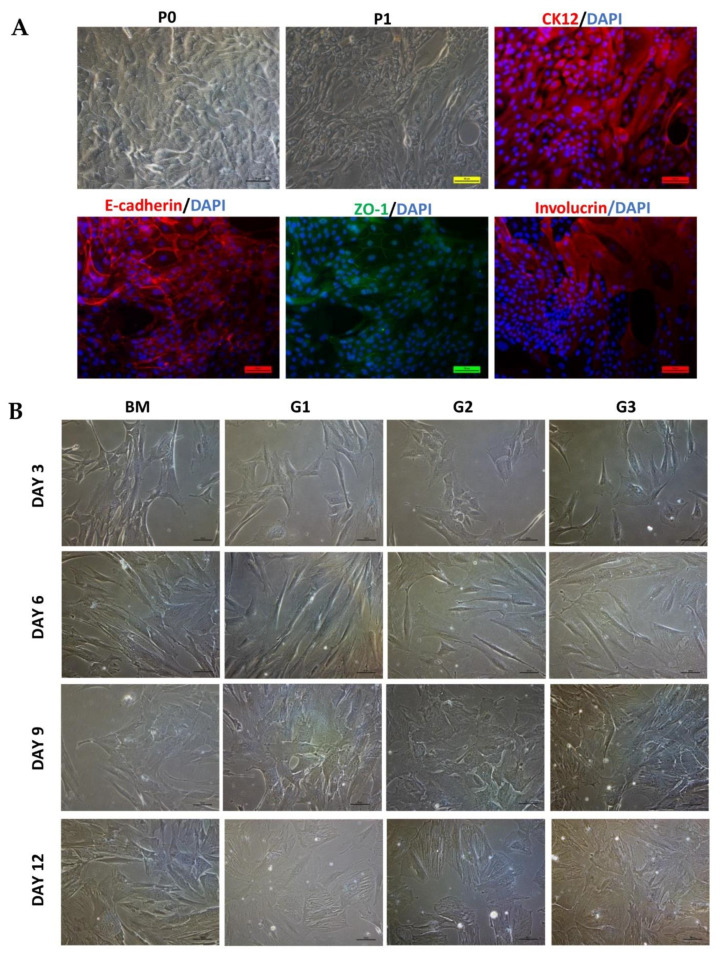
Morphology of CECs and WJ-MSCs during differentiation. (**A**) Morphology of CECs at Passage 0 (P0), P1 (bright field), and immunofluorescence stained with CK12, E-cadherin, ZO-1, Involucrin. (**B**) Morphology changed during differentiation into CECs from WJ-MSCs. Scale bar, 50 µm.

**Figure 5 ijms-23-03078-f005:**
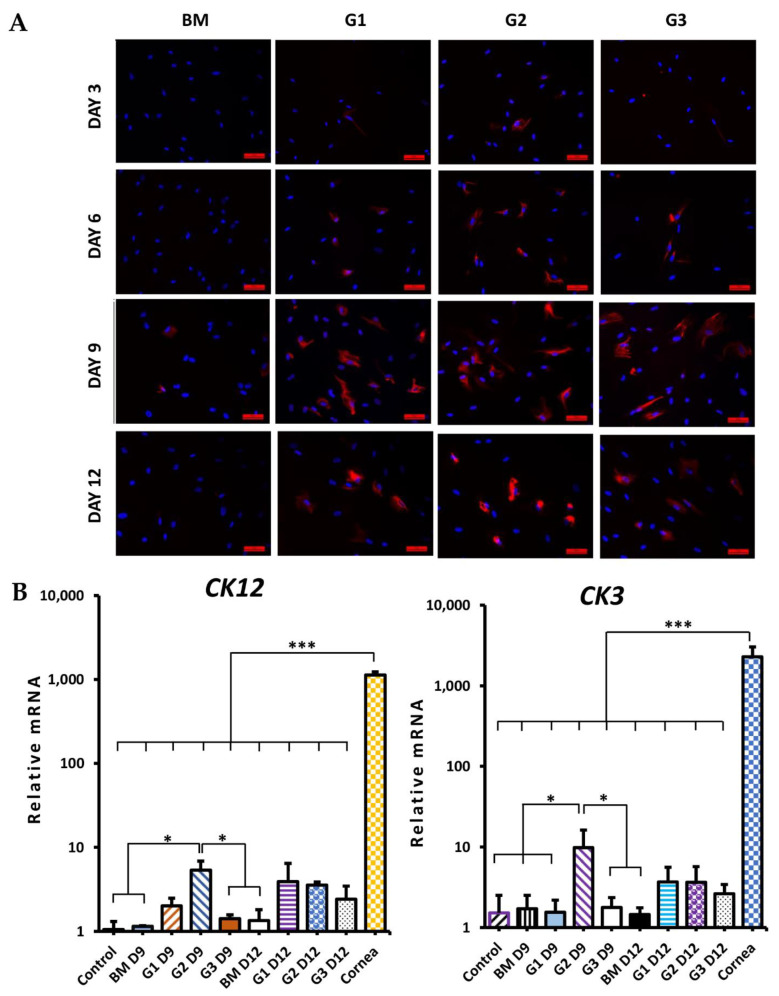
CK12, CK3 expression during CEC differentiation from WJ-MSCs. *(***A**) Immunofluorescence staining with CK12; red color (CK12), blue color (nucleus). Scale bar, 50 µm. *(***B**) The relative mRNA expression of CK12 (left) and the relative mRNA expression of CK3 (right). Data are presented as mean + SEM. * *p* < 0.05, *** *p* < 0.001.

**Figure 6 ijms-23-03078-f006:**
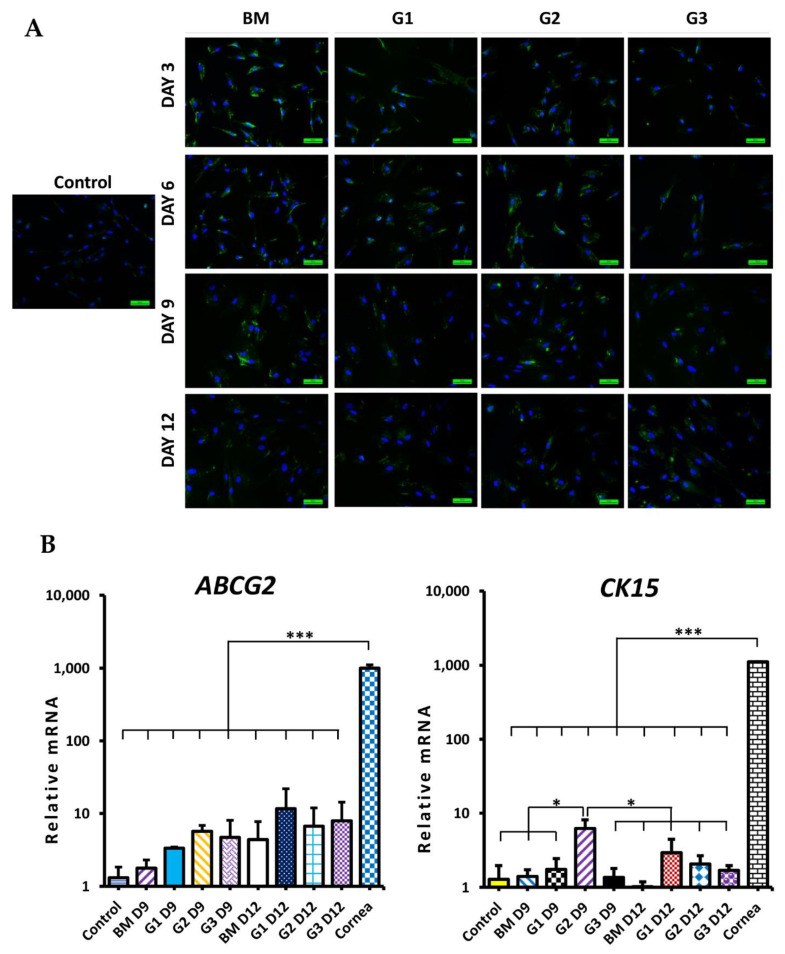
ABCG2, CK15 expression during CEC differentiation from WJ-MSCs. *(***A***)* Immunofluorescence staining with ABCG2; green color (ABCG2), blue color (nucleus). Scale bar, 50 µm (**B**) The relative mRNA expression of ABCG2 (left) and the relative mRNA expression of CK15 (right). Data are presented as mean + SEM. * *p* < 0.05, *** *p* < 0.001.

**Figure 7 ijms-23-03078-f007:**
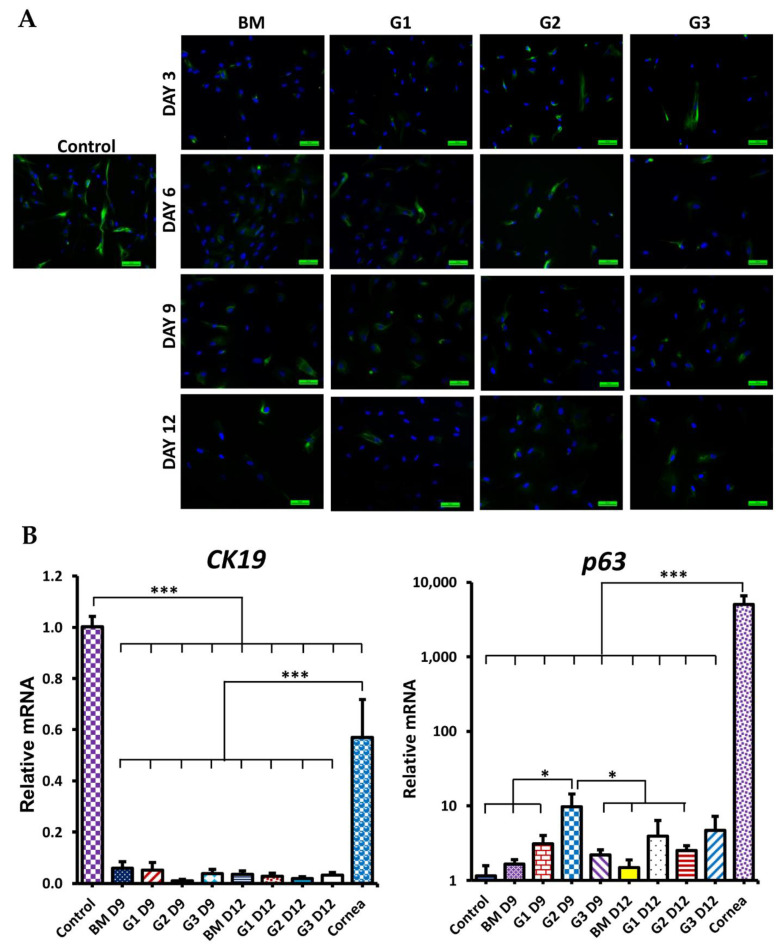
CK19, p63 expression during CEC differentiation from WJ-MSCs. (**A**) Immunofluorescence staining with CK19; green color (CK19), blue color (nucleus). Scale bar, 50 µm. (**B**) The relative mRNA expression of CK19 (left) and the relative mRNA expression of p63 (right). Data are presented as mean + SEM. * *p* < 0.05, *** *p* < 0.001.

**Figure 8 ijms-23-03078-f008:**
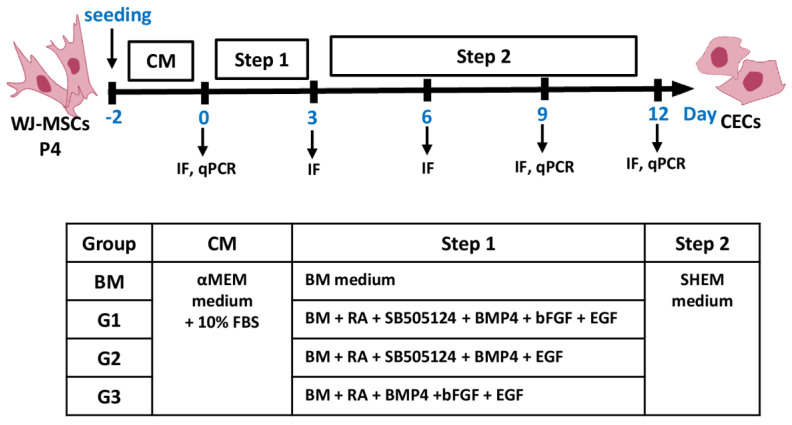
Schematic outline of CEC differentiation from human WJ-MSCs. IF: immunofluorescent.

## Data Availability

Data included in this study are available upon request from the corresponding author.

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
