# Peer review of "Signaling Pathways Impact on Induction of Corneal Epithelial-like Cells Derived from Human Wharton’s Jelly Mesenchymal Stem Cells"

_ijms, 2022, doi:10.3390/ijms23063078_

Round 1

Reviewer 1 Report

This is an interesting and well-designed study with sufficient methodology. However, there are a number of minor issues that the authors should address to be able to publish their results.

  1. Please correct in the legend of Figure 1A the “left” and “right” indication, since the pictures are up and down.
  2. Statistical significance should be denoted with asterisks in the figures.
  3. The authors should include in the legends what the error bars indicate.
  4. The discussion section needs improvement. The authors should discuss their results and further applications in treatment therapy with updated literature.

Author Response

Dear Sir/Madam,

Thank you very much for your careful revision.

We have carefully reviewed your comments and revised the manuscript accordingly. You can see in detail in the attached Response file.

Best regards,

Hong

Reviewer 2 Report

The Authors are presenting interesting results about the effects of treatments on three signaling pathways involved in CEC differentiation the definition of the most adequate protocol for inducing corneal epithelial differentiation from transplanted human WJ-MSCs. 

The scientific approach is correct and part of the results are in line with previous work by other Authors (please give more references). 

The introduction should be enriched by citing previous papers on this subject. This could give more emphasis to the novelty of their work. For example (but not only), papers like:

Garzon, I., Martin-Piedra, M.A., Alfonso-Rodriguez, C., Gonzalez-Andrades, M., Carriel, V., Martinez-Gomez, C., Campos, A., and Alaminos, M. Generation of a biomimetic human artificial cornea model using Wharton’s jelly me- senchymal stem cells. Invest ophthalmol vis sci 55, 4073, 2014.

Liu, H., Zhang, J., Liu, C.Y., Wang, I.J., Sieber, M., Chang, J., Jester, J.V., and Kao, W.W. Cell therapy of congenital corneal diseases with umbilical mesenchymal stem cells: lumican null mice. PLoS One 5, e10707, 2010.

Coulson-Thomas, V.J., Caterson, B., and Kao, W.W. Transplantation of human umbilical mesenchymal stem cells cures the corneal defects of mucopolysaccharidosis VII mice. Stem Cells 31, 2116, 2013.

Please reference and better underline the differences from their previous work  "Differentiation Induction of Human Stem Cells for Corneal Epithelial Regeneration. International Journal of Molecular Sciences 21:21, 7834".

Author Response

(The authors gave the same response as above.)

Round 2

Reviewer 2 Report

The Authors correctly revised the paper accordingly to my suggestions